# LEARNING UNDERLYING PHYSICAL PROPERTIES FROM OBSERVATIONS FOR TRAJECTORY PREDICTION

## ABSTRACT

In this work we present an approach that combines deep learning together with laws of Newton's physics for accurate trajectory predictions in physical games. Our model learns to estimate physical properties and forces that generated given observations, learns the relationships between available player's actions and estimated physical properties and uses these extracted forces for predictions. We show the advantages of using physical laws together with deep learning by evaluating it against two baseline models that automatically discover features from the data without such a knowledge. We evaluate our model abilities to extract physical properties and to generalize to unseen trajectories in two games with a shooting mechanism. We also evaluate our model capabilities to transfer learned knowledge from a 2D game for predictions in a 3D game with a similar physics. We show that by using physical laws together with deep learning we achieve a better human-interpretability of learned physical properties, transfer of knowledge to a game with similar physics and very accurate predictions for previously unseen data.

## 1 INTRODUCTION

Games that follow Newton's laws of physics despite being a relatively easy task for humans, remain to be a challenging task for artificially intelligent agents due to the requirements for an agent to understand underlying physical laws and relationships between available player's actions and their effect in the environment. In order to predict the trajectory of a physical object that was shot using some shooting mechanism, one needs to understand the relationship between initial force that was applied to the object by a mechanism and its initial velocity, have a knowledge of hidden physical forces of the environment such as gravity and be able to use basic physical laws for the prediction. Humans, have the ability to quickly learn such physical laws and properties of objects in a given physical task from pure observations and experience with the environment. In addition, humans tend to use previously learned knowledge in similar tasks. As was found by researchers in human psychology, humans can transfer previously acquired abilities and knowledge to a new task if the domain of the original learning task overlaps with the novel one (Council, 2000).

The problem of learning properties of the physical environment and its objects directly from observations and the problem of using previously acquired knowledge in a similar task are important to solve in AI as this is one of the basic abilities of human intelligence that humans learn during infancy (Baillargeon, 1995). Solving these two problems can bring AI research one step closer to achieving human-like or superhuman results in physical games.

In this work we explore one of the possible approaches to these two problems by proposing a model that is able to learn underlying physical properties of objects and forces of the environment directly from observations and use the extracted physical properties in order to build a relationships between available in-game variables and related physical forces. Furthermore, our model then uses learned physical knowledge in order to accurately predict unseen objects trajectories in games that follow Newtonian physics and contain some shooting mechanism. We also explore the ability of our model

to transfer learned knowledge by training a model in a 2D game and testing it in a 3D game that follows similar physics with no further training.

Our approach combines modern deep learning techniques (LeCun et al., 2015) and well-known physics laws that were discovered by physicists hundreds of years ago. We show that our model automatically learns underlying physical forces directly from the small amount of observations, learns the relationships between learned physical forces with available in-game variables and uses them for prediction of unseen object's trajectories. Moreover, we also show that our model allows us to easily transfer learned physical forces and knowledge to the game with similar task.

In order to evaluate our model abilities to infer physical properties from observations and to predict unseen trajectories, we use two different games that follow Newtonian Physics. The first game that we use as a testing environment for our model is Science Birds. Science Birds is a clone of Angry Birds - a popular video game where the objective is to destroy all green pigs by shooting birds from a slingshot. The game is proven to be difficult for artificially intelligent playing agents that use deep learning and many agents have failed to solve the game in the past (Renz et al., 2019). The second game that we are using as our testing environment is Basketball 3D shooter game. In this game the objective of a player is to shot a ball into a basket.

In order to test the abilities of our model to transfer knowledge to a different game we first train our model on a small amount of shot trajectories from Science Birds game and then test trained model for predictions of the ball trajectory in the Basketball 3D shooting game.

We compare the results of our proposed model that is augmented with physical laws against a two baseline models. The first baseline model learns to automatically extract features from observations without knowledge of physical laws, whereas the second baseline model learns to directly predict trajectories from the given in-game variables.

## 2  RELATED WORK

Previous AI work in predicting future dynamics of objects has involved using deep learning approaches such as: graph neural networks for prediction of interactions between objects and their future dynamics ( Battaglia et al. (2016); Watters et al. (2017); Sanchez-Gonzalez et al. (2018)), Bidirectional LSTM and Mixture Density network for Basketball Trajectory Prediction ( Zhao et al. (2017)) and Neural Physics Engine (Chang et al. (2016)). Some of the researchers also tried to combine actual physical laws with deep learning. In one of such works, researchers propose a model that learns physical properties of the objects from observations (Wu et al. (2016)). Another work proposes to integrate a physics engine together with deep learning to infer physical properties (Wu et al. (2015))

However, most of the work on predicting future objects dynamics is focused on learning physics from scratch or uses some known physical properties in order to train a model. This could be a problem as in most real-world physical games the underlying physical properties are not known to the player unless one has an access to the source code of a physics engine. Because of that, these properties have to be learned directly from experience with the environment without any supervision on the actual values of physical properties. Another important point, is that instead of learning physics from scratch we can use to our benefit already discovered and well-established laws of physics.

In this work we propose an approach that combines classical feedforward networks with well-known physical laws in order to guide our model learning process. By doing so, our model learns physical properties directly from observations without any direct supervision on actual values of these properties. Another contribution is that our model learns from a very small training dataset and generalizes well to the entire space. Furthermore, learned values can be easily interpreted by humans which allows us to use them in any other task in the presented test domains and can be easily transferred to other games with similar physics.

## 3 APPROACH

### 3.1 BASELINE MODELS

In order to measure the advantages of combining classical physical laws together with deep learning we are comparing our model against pure deep learning approaches with similar architectures.

#### 3.1.1 ENCODER BASELINE MODEL

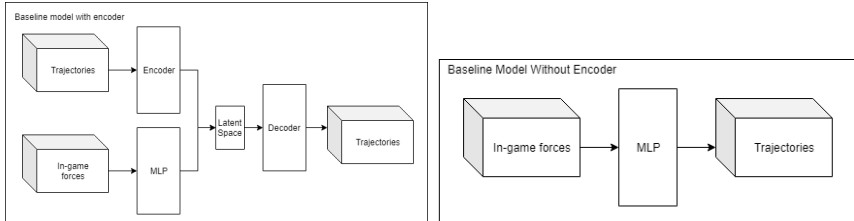

Figure 1: Left: First baseline model. The first neural network consists of encoder and decoder. Encoder takes in trajectories and encodes them into latent space. Decoder then takes the latent feature representation and decodes them back to trajectories. Loss is calculated using MSE and propagated back to both encoder and decoder. The second neural network receives in-game variables such as relative position of a bird or initial forces and learns to predict values predicted by encoder. These values then serve as input to the decoder which predicts the actual trajectories. For the prediction of unseen trajectories only the second neural network and the decoder are used. Right: Second baseline model. This model does not use observed trajectories and directly learns to predict the trajectories from initial forces or relative position of the physical object to the shooting mechanism.

Our first baseline model is based on the idea of autoencoders (Rumelhart et al. (1986)). Contrary to the proposed model in section 3.2 this model learns to automatically discover features from observations. It takes a sequence of points $T = \{(x_0, y_0), (x_1, y_1), ..., (x_n, y_n)\}$ as its input and encodes it to a latent space $T_{enc}$. The encoded trajectory is then used by decoder to reconstruct the trajectory. The second part of this baseline model consists of another MLP that learns to associate a relative position of a physical object that generated the trajectory with learned latent space $T_{enc}$. More formally, given trajectory $T$ as input, encoder $f_{encoder}$, and decoder $f_{decoder}$, our model reconstructs a trajectory $\hat{T}$ from latent space as follows:

$$\hat{T} = f_{decoder}(f_{encoder}(\{(x_0, y_0), (x_1, y_1), ..., (x_n, y_n)\})) \tag{1}$$

Once the trajectory is reconstructed, we compute the loss using Mean Squared Error and update the weights of our networks:

$$\frac{1}{n} \sum_{i=1}^{n} (T - \hat{T})^2 \tag{2}$$

The second part of this baseline model is a another MLP $f_{associate}$ that learns to associate given initial relative position of a physical object $(x_{r_0}, y_{r_0})$ with derived in a previous step encoded trajectory $T_{enc}$:

$$\hat{T_{enc}} = f_{associate}((x_{r_0}, y_{r_0})) \tag{3}$$

In order to update the weights of $f_{associate}$ we compute the loss using Mean Squared Error between two derived encodings $T_{enc}$ and $\hat{T_{enc}}$.

After that, we predict trajectory using derived $\hat{T_{enc}}$ as follows: $\hat{T} = f_{decoder}(\hat{T_{enc}})$

#### 3.1.2 SIMPLE BASELINE MODEL

In order to evaluate the advantages of using observations and an encoder-decoder scheme, we use the second baseline model that does not use encoder-decoder and directly learns to predict trajectory

from the given in-game forces or relative position of a physical object. More formally, given relative position of a physical object $(x_{r_0}, y_{r_0})$ and MLP $f_{simple}$, we compute the trajectory as follows:

$$\hat{T} = f_{simple}((x_{r_0}, y_{r_0})) \tag{4}$$

## 3.2 PHYSICS AWARE MODEL

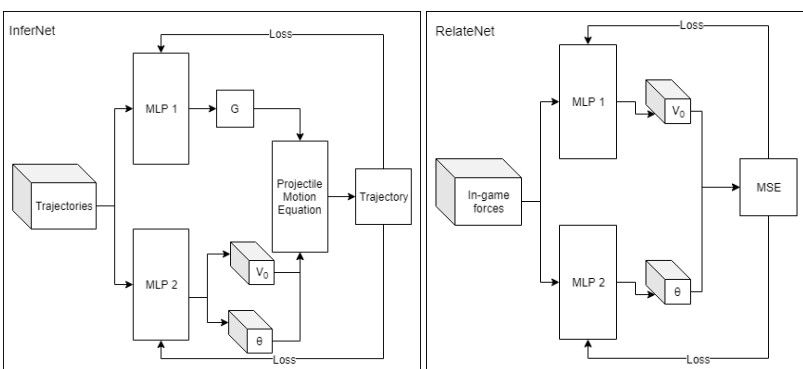

Figure 2: Left: InferNet internal architecture. MLP 1 predicts a single value that represents a gravity, while MLP 2 predicts a batch of values that represent initial velocities $V_0$ and angles $\alpha$ for each trajectory in the input batch. These values are then inserted to the projectile motion equation to calculate the trajectory of the object. The predicted trajectory is then used to calculate the loss and update weights of both networks. Right: RelateNet internal architecture. MLP 1 predicts a batch of values that represent initial velocities $V_0$, while MLP 2 predicts a batch of values that represent angles $\alpha$ for each trajectory in the input batch. The predicted values are then used to calculate the loss and update the weights of both networks.

Similarly to the first baseline model presented in section 3.1.1, Physics Aware Network (PhysANet) consists of two parts: a neural network that discovers physical forces of the environment and action that generated given observations and a neural network that learns the relationship between the in-game actions or forces and predicted physical values. We further refer to these two parts as InferNet and RelateNet.

### 3.2.1 INFERNET

The goal of InferNet is to extract physical forces that have generated a given trajectory $\{(x_0, y_0), (x_1, y_1)...(x_n, y_n)\}$ using guidance from known physical equations. The discovered physical forces are then plugged to the projectile motion equation in order to calculate the trajectory.

InferNet consists of two internal small MLPs that are trained together as shown on Figure 2 (Left). The first MLP takes in a batch of trajectories and predicts a single value of gravity force for all of them. The second MLP takes in a batch of trajectories and for each trajectory in a batch it predicts an initial velocity and angle of a shot. These predicted values are then inserted into a projectile motion equation in order to calculate the resulting trajectory. The projectile motion equation is defined as follows (Walker (2010)):

$$y = h + x \tan(\theta) - \frac{gx^2}{2V_0^2 cos(\theta)^2} \tag{5}$$

In equation 5, $h$ is the initial height, $g$ is gravity, $\theta$ is the angle of a shot and $V_0$ is initial velocity.

Once it had calculated the trajectory we compute the loss between observed trajectory $T$ and predicted trajectory $\hat{T}$ using Mean Squared Error in a similar way as was defined in equation 2.

### 3.2.2 RELATENET

The goal of RelateNet that is shown on Figure 2 (Right) is to learn the relationship between in-game variables and physics forces predicted by InferNet. This network tries to predict extracted by InferNet forces directly from the given in-game values such as relative position of a bird. The in-game variables can be any variables with continuous or discrete domain that can be chosen by the playing agent in order to make a shot. As an example, in-game variables can be the initial forces of the shot or object's relative position to the shooting mechanism. RelateNet consists of two internal MLPs where first MLP predicts initial velocities and second MLP predicts initial angles. In order to update the weights of both internal MLPs, we calculate the MSE between values predicted by InferNet and values predicted by RelateNet. More details on the architecture of the PhysANet can be found in the Appendix A.

## 4 EXPERIMENTS

In our experiments we are using Science Birds (Ferreira, 2018) and Basketball 3D shooter game (Nagori, 2017) as a testing environments.

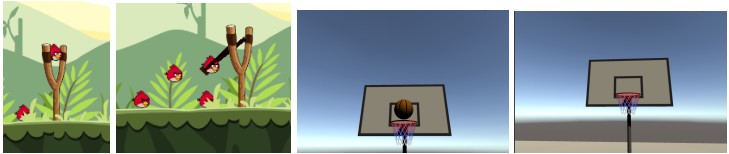

Figure 3: Slingshot in Science Birds game and Basketball 3D shooter game.

### 4.1 SCIENCE BIRDS TRAJECTORY PREDICTION

Science Birds is a clone of Angry Birds which has been one of the most popular video games for a period of several years. The main goal of the game is to kill all green pigs on the level together with applying as much damage as possible to the surrounding structures. The player is provided with a sequence of (sometimes) different birds to shoot from a slingshot. In Science Birds, similarly to Angry Birds all game objects are following the laws of Newton's Physics (in 2D). In order to predict the outcomes of the shots, the player needs to understand basic physical laws that govern the game and be able to use them for a prediction of a bird's trajectory.

Science Birds contains a slingshot (Figure 3) which serves as a shooting mechanism that allows a player to shoot a bird. Player can select a strength and the angle of the shot by changing a relative position of the bird to the center of a slingshot. In order to accurately predict the trajectory of a bird, player needs to have an understanding of the relationship between relative position of the bird to the slingshot and resulting trajectory. The underlying physical properties of the game such as gravity, or initial velocity of a related shot or its angle is unknown to the player and has to be estimated from the observations and experiences.

The goal of our model in this experiment is to learn the relationship between relative position of a bird and initial velocity and angle of the resulting shot in order to predict trajectories of previously unseen shots. In this experiment we do not provide actual physical properties or forces to our model and it has to learn them directly from observations.

In order to train our model for this experiment, we have collected a small data set of 365 trajectories and relative positions of the bird that generated these trajectories from Science Birds. This data set was then split to train, test and validation data sets, resulting in a training data set that contains only 200 trajectories. We are using such small training data set in order to prevent a model to simply memorize all possible trajectories and instead learn to use extracted physical forces and properties to predict unseen trajectories.

During the training of a model, PhysANet takes in a trajectory as input and estimates the angle $\theta$, initial velocity $V_0$ and gravity $g$ that generated this trajectory. The predicted values are then inserted to projectile motion equation (5) in order to recreate the trajectory. As a second step, the relative

position of the bird to the slingshot $(x_0, y_0)$ is fed as input to the RelateNet which learns to predict values $\theta$ and $V_0$ predicted by InferNet.

During the testing of a model, we only feed the relative position of the bird to RelateNet which predicts values $\theta$ and $V_0$ that are plugged in to projectile motion equation in order to calculate the trajectory.

## 4.2 Basketball 3D Shooter Game

Basketball 3D shooter is a game where the player has to throw a ball to the basket in order to earn points. In this game the shooting mechanism is different compared to Science Birds and the initial force that is applied to the physical object does not depend on the relative position of that object.

In this experiment we are interested in the ability of our model to transfer the learned relationships and physical properties from one game with shooting mechanism to the other. In particular, we were interested in ability of our model to use learned knowledge from Science Birds for predictions of the ball's trajectory in Basketball 3D Shooter. Because of the absence of the slingshot like mechanism in this game, we train our models to predict the trajectories based on the initial force that is applied to the bird or a ball by a shooting mechanism when launched.

As in the Science Birds experiment described in section 4.1, the input to the InferNet is a trajectory $\{(x_0, y_0), (x_1, y_1)...(x_n, y_n)\}$ and an input to the RelateNet is initial force that is applied to the bird or a basketball ball by a shooting mechanism. We note here that the magnitude of the initial force applied to the ball in Basketball 3D Shooter game is higher than the magnitude of the initial force applied to the bird in Science Birds.

## 4.3 Testing datasets

In order to evaluate our model we are using three testing datasets: Test, Generalization and Basketball. Test dataset contains trajectories generated by previously unseen (by the model) initial forces or relative positions of a bird. Generalization dataset contains trajectories of the shots made to the opposite direction of shots in the training set. In particular, it contains trajectories and the related relative positions of the bird of shots to the left side of a slingshot. The basketball dataset contains trajectories and related to them initial forces that were applied to the ball in the Basketball 3D shooter game.

These three datasets are not exposed to our models during the training. In the Basketball experiment described in section 4.2, we do not train our models on the basketball dataset but only on the training dataset from Science Birds.

# 5 Results

## 5.1 Prediction of trajectories in Science Birds

The results of our first experiment show that PhysANet has the most accurate trajectory prediction out of all tested models. PhysANet was able to learn to associate a position of a bird relative to the slingshot with initial velocity and angle of a shot as shown on Figure 4. In particular Figure 4 shows two learned relationships between position of a bird before it is shot from a slingshot and initial parameters of the shot. The graph on the left side of Figure 4 shows how the initial angle between bird and center of a slingshot affects the initial angle of a trajectory. As an example from this graph we can see that bird positioned at angle of 220 degrees would result in trajectory that has initial angle of roughly 45 degrees which as we know from a trigonometry is close to the "truth". The graph on the right side of Figure 4 shows how initial velocity depends on the distance between initial bird position and center of the slingshot. In order to show this relationship we have fixed the bird at an angle of 225 relatively to the center of slingshot and only changed the distance. From this graph we can see that as one could expect the more we extend the slingshot the stronger our shot is going to be and the higher initial velocity will be.

In Figure 5 we present a few examples of the trajectory predicted by PhysANet and actual trajectories that bird took. The presented examples show three trajectories from a test dataset which the network

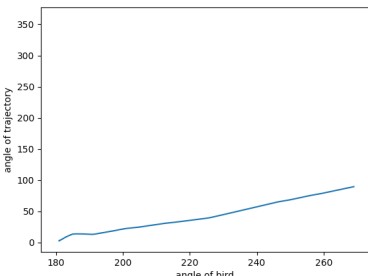 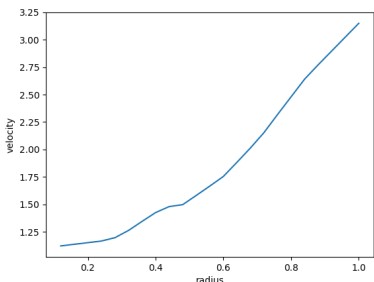

Figure 4: Learned relationships between velocities and angle and relative position of a bird. Graph on the left side shows how initial angle between bird and center of a slingshot related to the initial angle of the trajectory. Graph on the right side shows how initial velocity is changing when increasing the distance between bird and center of the slingshot. In this example angle between bird and slingshot was fixed at 225 degrees.

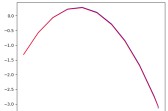 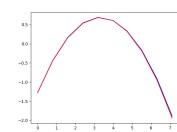 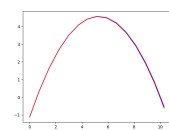 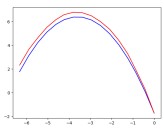

Figure 5: Predicted and actual trajectories of a bird. Red line represent a predicted trajectory by PhysANet, while blue line represent a true trajectory taken by the bird. The first three trajectories are from test data set, whereas the last one is from generalization dataset.

had never seen before and one trajectory of the shot to the left side of a slingshot. Despite the fact that our training dataset did not contain the shots to the left side of the slingshot, our model shows a good generalization ability and predicts the trajectory quite accurately. This shows that learned physical properties and relationships between position of a bird and slingshot can be used by PhysANet to accurately predict the trajectories.

As shown in Table 1, PhysANet showed significantly better overall accuracy on test and generalization datasets than our two baseline models.

## 5.2 TRANSFER OF KNOWLEDGE TO BASKETBALL 3D

The results in this experiment show that learned physical properties and relationships in Science Birds can be transferred to the Basketball 3D shooter game. Due to the fact that model was trained on data from a 2D environment, we tested the predictions of 3D trajectories by separating prediction process into predictions in x,y and z,y planes. Surprisingly, despite being trained in a 2D environment, PhysANet showed good results of predicting the trajectories in both planes without any additional training in new 3D environment.

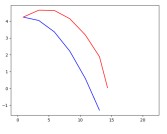 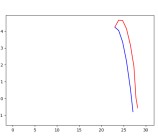 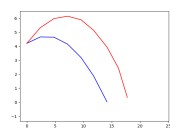 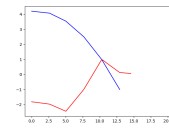 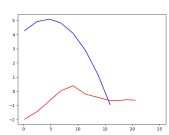

Figure 6: Comparison of model's abilities to transfer learned knowledge to Basketball 3D. The first two pictures show predictions of one trajectory in x,y and z,y planes respectively. The rest of the pictures show predictions only in x,y plane. The first three pictures on the left show trajectories predicted by PhysANet, fourth picture shows a trajectory predicted by BM1, the last picture shows a trajectory predicted by BM2.

Table 1: Average observed Mean Squared Error between actual trajectory and predicted trajectory.

| DATASET | PhysANet MSE | BM1 MSE | BM2 MSE |
|---|---|---|---|
| Test | **0.324** | 0.554 | 0.832 |
| Generalization | **0.837** | 1.637 | 2.162 |
| Basketball 3D | **4.051** | 8.196 | 7.676 |

The Figure 6 shows the examples of predictions of trajectories from the Basketball dataset by PhysANet, BM1 and BM2. The first three pictures show trajectories predicted by PhysANet, fourth picture shows prediction made by BM1 and the last picture shows prediction made by BM2. The first two pictures show predictions for one trajectory in x,y and z,y planes. The rest of the pictures shown in Figure 6 show predictions in x,y plane only. As we can see from Figure 6, despite the fact that initial forces in Basketball 3D shooter game have higher magnitudes, the PhysANet was able to correctly handle it and still predict trajectories in a correct shape and have a relatively low accuracy error. This is a contrary to the two baseline models that could not handle new environment properly and predicted seemingly random trajectories despite being relatively accurate in the predictions in the first experiment described in section 4.

The results presented in Table 1 show the comparison of the observed Mean Squared Errors of all three models. These results show that PhysANet surpasses two baseline models in all testing datasets. Thus for example, error shown by PhysANet in Basketball dataset is nearly two times lower than error shown by Baseline Model 1 (BM1) which was described in section 3.1.1. Surprisingly, despite the fact that BM1 showed better results in Test and Generalization datasets than BM2, it showed higher prediction error in the Basketball 3D dataset than a simple baseline model (BM2). Despite the fact that PhysANet showed relatively good results in predicting trajectories in previously unseen game there still seems to be some loss in the accuracy of the predictions. We hypothesise that the loss of the accuracy can be caused by the differences in physics engines of the two games. One of the possible solutions of this problem is to improve our model abilities to adapt to the new environment after observing a few shots, however we leave such an improvement for a future research.

## 6 DISCUSSION

In this work we have showed that combining known physical laws with deep learning can be beneficial for learning more human-interpretable and flexible physical properties directly from the small amount of the observations. We showed that a model that is augmented with physical laws achieved better generalization to unseen trajectories, was able to transfer learned knowledge to a different game with similar physics and made generally more accurate predictions of trajectories than models without such physical knowledge. Our results show that in some situations using already discovered physical laws and equations can be beneficial and can guide deep learning model to learn physical properties and relationships that lead to more accurate predictions than predictions made by models that learned features automatically without any supervision. Because learned values can be easily interpreted by a human these values can be used in potentially new tasks. As an example, learned gravity of the environment can be used for prediction of a physical behaviour of other physical objects in the environment. Another important quality of our approach is that learned knowledge can be transferred to a task with a similar physics as was shown by the Basketball experiment. Lastly, our model can be easily adopted to any other physical task by changing or adding more physical equations and learning different physical properties from the observations. This potentially could allow us to fully master the physics of a game and use learned knowledge as part of artificially intelligent agent in order to achieve human-like performance in physical games.

Our results show that it could be beneficial to use physical laws that were discovered by physicists hundreds of years ago together with deep learning techniques in order to unleash their full predicting power and bring artificially intelligent agents a step closer to achieving human-like physical reasoning.

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

## A    APPENDIX

## B    PHYSANET ARCHITECTURE

As was mentioned in section 3.2, PhysANet has two separate neural networks: InferNet and RelateNet.

InferNet consists of two neural networks (MLP1 and MLP2) each with a single hidden layer of size 200 and 100 respectively. MLP1 takes in trajectories consisting of at most 25 points. Trajectories

with less than 25 points are padded with zeros, and trajectories with more than 25 points are cut. Trajectories and related in-game variables are fed to the network in batches of size 8.

RelateNet also consists of two neural networks (MLP 1 and MLP 2) each with a single hidden layer of size 100. MLP 1 and MLP 2 both take in the in-game variables in batches of size 8.

For all hidden layers we are using ReLU activation function and no activation function for the last layer.

In order to train PhysANet, we first train InferNet until it converges to value of a loss being close to zero and after that we train RelateNet. Simultaneous training of both networks is also possible, however in our experiments training RelateNet after InferNet showed better results.

