# OpenReview forum: "Learning Underlying Physical Properties From Observations For Trajectory Prediction"
_ICLR.cc/2020/Conference — Reject_

### Official Review · AnonReviewer3 · 2019-10-17
**Official Blind Review #3**

**Rating:** 1

**Review:**

The problem addressed by this paper is the estimation of trajectories of moving objects thrown / launched by a user, in particular in computer games like angry birds or basketball simulation games. A deep neural network is trained on a small dataset of ~ 300 trajectories and estimates the underlying physical properties of the trajectory (initial position, direction and strength of initial force etc.). A new variant of deep network is introduced, which is based on an encoder-decoder model, the decoder being a fully handcrafted module using known physics (projectile motion).

I have several objections, which can be summarized by the simplicity of the task (parabolic trajectories without any object/object or object/environment collisions / interactions), the interest of the task for the community (how does this generalize to other problems?), and the writing and structuring of the paper. I will detail these objections further in the rest of the review.

Learning physical interactions is a problem which has received considerable attention in the computer vision and ML communities. The problem is certainly interesting, but I think we should be clear on what kind of scientific knowledge we want to gain by studying a certain problem and by proposing solutions. The tasks studied by the community are mostly quite complex physical phenomena including multiple objects of different shapes and properties and which interact with each other. All these phenomena can be simulated with almost arbitrary precision with physics engines, and these engines are mostly also used for generating the data. In other words, the simulation itself is solved and is not the goal of this body of work. The goal is to learn differentiable models, which can be used as inductive bias in larger models targeting more general tasks in AI.

Compared to this goal, the proposed goal is far too easy: learning projectile motion is very easy, as these trajectories can be described by simple functions with a small number of parameters, which also have a clear and interpretable meaning. The simplicity of the task is also further corroborated by the small number of samples used to estimate these parameters (in the order of 300). A further indication is the fact, that the decoder in the model is fully hardcoded. No noise modelling was even necessary, which further corroborates that a very simple problems is addressed.

In order words, I am not really sure what kind of scientific problem is solved by this work, and how this knowledge can help us to solve other problems, harder problems.

My second objection is with the written form of the paper. The paper is not well enough structured and written, many things are left unsaid. First of all, the problem has never been formally introduced, we don’t know exactly what needs to be estimated. What are the inputs, outputs? Is computer vision used anywhere? How are the positions of the objects determined if not with computer vision? How are the user forces gathered? What are “in game variables” mentioned multiple times in the document? No notation has been introduced, no symbols have been introduced (or too late in the document). For instance, there is no notation for the latent space of the encoder-decoder model.

The figures are not very helpful, as the labelling of the blocks and labels is very fuzzy. As an example, For InferNet, inputs and trajectories are “Trajectories”, so what is the difference? Of course we can guess that (inputs are measured trajectories, outputs are reconstructed trajectories), but we should not guess things when reading papers.

The figure for encoder-decoder model is very confusing, as the different arrows have different functional meanings and we have no idea what they mean. The outputs of the encoder and the MLP both point to the latent space and at a first glance the reader might think that they are concatenated, which raises several questions. Reading the text, we infer that first a model is trained using on one of the arrows (the one coming from the encoder) and ignoring the other one, and then the MLP is learned to reconstruct the latent space using the other arrow (the one coming from the MLP), but this is absolutely impossible to understand looking at the figure, which does not make much sense. We can infer all this from the text around equations (1) to (3), which is itself quite fuzzy and difficult to understand, in spite of the simplicity of the underlying maths.

The relationship of RelateNet and InferNet is not clear. While the task of InferNet is clear, the role of InferNet in the underlying problem is not clear and it has not been described how it interacts with RelateNet.

It is unclear how the transfer between science birds and basketball has been performed and what exactly has been done there.

As mentioned above, the role of “in game variables” is unclear. What are those? I suggest to more clearly define their roles early in the document and use terms from well-defined fields like control (are they “control inputs”) or HCI (are they “user actions”?).

In the evaluation section, we have baseline models BM1 and BM2, but they have never been introduced. We need to guess which of the models described in the paper correspond to these.

The related work section is very short and mostly consists of an enumeration of references. The work should be properly described and related to the proposed work. How does the proposed work address topics which have not yet been solved by existing work?


**Experience Assessment:**

I have published in this field for several years.

**Review Assessment: Checking Correctness Of Derivations And Theory:**

I carefully checked the derivations and theory.

**Review Assessment: Checking Correctness Of Experiments:**

I carefully checked the experiments.

**Review Assessment: Thoroughness In Paper Reading:**

I read the paper thoroughly.

---

> ### Author Response · Authors · 2019-11-12
> **Response to reviewer #3**
>
> We would like to thank the reviewer for their review and comments. We have taken into account remarks on the figures and the structure of the paper and will fix them in our revision.
>
>
> “In order words, I am not really sure what kind of scientific problem is solved by this work, and how this knowledge can help us to solve other problems, harder problems.”
>
>
> One of the goals of our paper was to design a network that would learn to predict the trajectory of a shot in games that follow Newton’s Physics when it has no direct access to the physics engine of the game. As an example task, we picked Science Birds - a clone of a popular game Angry Birds which requires player to shoot a bird from a slingshot. Predicting a trajectory directly from observations in this case is a non trivial task as it requires the agent to understand the relationship between the relative position of a bird and shot trajectory. Our method then can be combined with playing agents that would use trajectory prediction in making decisions. This is actually one of the problems agents face in the Angry Birds AI competition where they only get screenshots of the game but don’t have access to the physics engine and don’t know the physics parameters.
>
>
> Similar methods can be used for many other situations/problems where we know the physics equations that apply, but don’t know the required physics parameters of objects and the environment. As such, we believe that our work is important for solving these kinds of problems and also an important new approach. In our work we showed that rather than learning already known physics equations, we can learn the required physics parameters from the observations and use them successfully in predictions.

---

> > ### Comment · AnonReviewer3 · 2019-11-14
> > **After rebuttal**
> >
> > Thank you for this reponse, but it addresses only a single one of my questions, and it is not really an answer to my concerns. No other question was addressed.
> >
> > I will keep my rating.

---

### Official Review · AnonReviewer1 · 2019-10-17
**Official Blind Review #1**

**Rating:** 3

**Review:**

This paper presents a method for predicting the trajectories of objects in video (or phone) games by learning the physical parameters underlying the movements. To do so, the authors used some multi-layer perceptrons on the (x,y) trajectory in order to i) estimate the physical values of the equation (equation supposed to be known, here a parabolic trajectory); and then ii) predict the trajectories from new initial conditions.

General comment:
While the links between physics and machine learning is clearly interesting and trendy, I found the paper unclear, not well motivated and I think the work is not enough for a paper in ICLR. Yet, I think the authors did spend some time and this work might be suited for a workshop.

Positive aspects:
- the authors really tried to focus on games that are used today.
- the results are showing that they do learn the parameters they wanted, as the new trajectories are indeed working.

Remarks and questions:
- the writing of the paper is not enough to make it clear, and a lot of sentences are not readable. Starting at the second sentence of the abstract. Try to make small sentences, and add all the pronouns 'a', 'the', .... Avoid the too numerous 'and' and cut the sentences.
- What is a shooting mechanism? Is it something scientifically defined? otherwise, explain better: the parabolic trajectory is I guess more explanatory of what you are doing.
- What is the goal of your work? Estimating the parameters of an equation used in a game is not really interesting.. as we have to know the equation, it has to be simple, we have to extract the trajectory from the game... but there might be other related applications that could motivate this work.
- Related work: I don't really see the novelty of your work. You are saying that the physical properties have to be learned from experience, but you are actually relying on a known equation, and just tuning the 3 parameters. Do games always follow the physical laws? From my knowledge, some games change the physical laws in order to be more pretty, ect. in that case, your method will not work?
- More generally, can you explain the difference for you between physical laws and physical properties?
- Explain what is 'MLP'; how many layers, what size, how did you defined it, ect.
- in 3.1.1. you are spending quite a long time in explaining what an autoencoder is; I think you can go faster on this.
- Why the RelateNet has 2 distinct MLPs while InferNet uses the same for inferring V0 and Theta?
- How did you extract the trajectories, ect. from the games?
- Figure 6: How come in the two last images, we see different starting points?
- Table 1: can we have an idea of the errors in meters? and compared to the distance of the trajectory?

Small remarks:
- theta is not alpha. Please use a common notation.
- Figure 5, 6.. : the legends are clearly not visible. they might not be useful, but in this case you have to spend time in changing the tick labels so that we can read them.
- 'build a relationships': no 's'
... a lot of  grammatical/ sentence problems



**Experience Assessment:**

I have read many papers in this area.

**Review Assessment: Checking Correctness Of Derivations And Theory:**

I carefully checked the derivations and theory.

**Review Assessment: Checking Correctness Of Experiments:**

I assessed the sensibility of the experiments.

**Review Assessment: Thoroughness In Paper Reading:**

I read the paper at least twice and used my best judgement in assessing the paper.

---

> ### Author Response · Authors · 2019-11-12
> **Response to reviewer #1**
>
> We would like to thank the reviewer for their remarks and comments. We would like to provide an answer to the questions raised in your review:
>
> "- What is a shooting mechanism? Is it something scientifically defined? otherwise, explain better: the parabolic trajectory is I guess more explanatory of what you are doing."
>
> Shooting mechanism can be defined as any game object that launches another game object with some initial velocity and can be controlled by a player.
>
> “- What is the goal of your work? Estimating the parameters of an equation used in a game is not really interesting.. as we have to know the equation, it has to be simple, we have to extract the trajectory from the game... but there might be other related applications that could motivate this work.”
>
>
> As described in the comment for reviewer #4, estimating the parameters despite knowing the equation is the essential task we face whenever we want to predict consequences of physical actions. The goal of our work is to learn to predict the trajectories of the shot given the observations. In order to achieve it, we have designed a network that learns physical properties from the observed trajectories without any insight on the values of these properties. In the concrete example we consider, such a module could then be combined with playing agents in order to play games that follow Newton’s physics and have a shooting mechanism that “creates” trajectories.
>
> "- Do games always follow the physical laws? From my knowledge, some games change the physical laws in order to be more pretty, ect. in that case, your method will not work?"
>
> In this work we are focusing on solving the trajectory prediction task in games that do follow Newton's laws of motion, the general solution for the cases where these laws do not apply is not a focus of this paper and is left to the future work.
>
> "- More generally, can you explain the difference for you between physical laws and physical properties?"
>
> Physical properties is something that can be measured by observation, physical law is something that was derived by scientific experiments and can be used to predict physical behavior if certain preconditions apply.
>
> "- Explain what is 'MLP'; how many layers, what size, how did you defined it, ect."
>
> 'MLP' or in another words multilayered perceptron is a feedforward artificial neural network. We provide the details on architecture of the used MLPs in Appendix.
>
> "- Why the RelateNet has 2 distinct MLPs while InferNet uses the same for inferring V0 and Theta?"
>
> By our experiments we have determined that using 2 distinct MLPs in RelateNet showed better results than using a single MLP as in InferNet.
>
> "- How did you extract the trajectories, ect. from the games?"
>
> The trajectories of objects were extracted directly from the physics engine of the game. In order to do so, we were tracking launched object throughout time starting from the moment in which the object was launched and until the moment in which object hits the ground. At each time step we were recording the position of a the launched object which resulted in a sequence of points. The resulted sequences were then padded with zeros or cut to the desired size.
>
> "- Figure 6: How come in the two last images, we see different starting points?"
>
> The goal of the Baseline Models 1 and 2 was to predict the trajectory of the launched object. The last two images on Figure 6 demonstrate the inability of the two baseline models to predict the correct position of the object on y-axis (when applied to the Basketball dataset with no further training) even at the first time step (starting point in the graph).

---

### Official Review · AnonReviewer4 · 2019-11-04
**Official Blind Review #4**

**Rating:** 3

**Review:**

This paper proposes an architecture that encodes a known physics motion equation of a trajectory of a moving object. The modeled equation has 3 variables and the network works in a latent space- contrary to taking raw images. It uses an auxiliary network (named InferNet) to train the final one used at inference time (named RelateNet). The former aims to reconstruct the input sequence of positions representing trajectory, and has intermediate 3 latent variables that correspond to the 3 variables of the modeled equation, while as decoder it uses the modeled known equation itself. The latter is a mapping from the relative position of the object to 2 latent variables of the former InferNet, and is trained with MSE loss. At inference, RelateNet takes as input the relative position of the object, predicts 2 variables of the equation and finally uses the motion equation to calculate the trajectory.

It is not easy for me to understand the use-case of the proposed method. In which real-world scenarios we would have the exact motion equation, and why given that we know such an equation we would want to learn a mapping from the relative position to a trajectory. In other words, it would be much more useful to learn the projectile motion equation itself. How does the proposed method handle input sequences which do not follow equation 5? To use this method do we need to know in advance the exact motion equation and its relevant ‘in-game variables’? In which cases would the former hold and in which cases would the latter be easy to obtain from raw pixels? Could the authors elaborate on it?

If I understand correctly, the trajectories (the input to InferNet) were generated with known $G$, $V_0$ and $\theta$ (the 3 latent variables of InferNet). It is not clear to me why the authors don’t use these for the MSE loss used to train InferNet (rather than using the projectile motion equation).

In my opinion, the introduction and related work sections do not reflect what is proposed in the paper. As an example, paragraph 2 of the introduction refers to use-cases where we would like to learn dynamics that govern certain motions directly from observations, whereas the proposed method uses extracted positions as input, and handcrafts the motion equation. The third paragraph of page 2 mentions agents failing to solve a game with Newtonian physics, whereas the method in this paper does not demonstrate empirically a way that this architecture could be used by an agent.

- Is the ‘projectile motion equation’ missing from Fig.2-right; is it used for inference? Is G from InferNet also input to RelateNet?

In summary, in my opinion, the technical novelty of this paper is limited as it uses MLP mappings that in some sense aim at learning the inverse of the equation that generated the data. Moreover, after reading the paper the use-case of the proposed method is not clear to me and the writing is unclear (see examples above and below).

— Minor —
- The term ‘in-game variables’ is used in a few places and is explained later in the text (Pg.5). I think that It would be helpful if it is explained in more detail the first time it is mentioned.
- I don’t understand the second sentence of the abstract.
- Pg1: build a relationships -> build relationships.
- Pg2: I don’t understand what the authors mean by ‘clone of Angry Birds.’
- Pg3: is $f_{associate}$ trained jointly or afterwards?
- Pg4: was MSE the loss used for $f_{simple}$?
- It would help adding sub-captions in Fig. 6.


**Experience Assessment:**

I have published one or two papers in this area.

**Review Assessment: Checking Correctness Of Derivations And Theory:**

N/A

**Review Assessment: Checking Correctness Of Experiments:**

I assessed the sensibility of the experiments.

**Review Assessment: Thoroughness In Paper Reading:**

I read the paper thoroughly.

---

> ### Author Response · Authors · 2019-11-12
> **Response to reviewer #4**
>
> We would like to thank the reviewer for their constructive comments and review. We would like to provide an answer to the questions raised in your review:
>
> “It is not easy for me to understand the use-case of the proposed method. In which real-world scenarios we would have the exact motion equation, and why given that we know such an equation we would want to learn a mapping from the relative position to a trajectory”
>
>
> The use case of our method is the following: We know physics and we know the underlying physics formulas that apply when observing a physical action. What we don’t know is the required physics parameters of objects and the environment we observe, for example mass, friction, temperature, air pressure, air resistance, gravity, etc. So despite knowing the physics formulas that apply, we cannot predict consequences of actions.
>
>
> In the example we use in the paper, a playing agent has to predict the trajectory of the shot in a game that follows Newtonian Physics. The agent only has access to the images of the game and no access to the physics engine, that is, the agent knows the underlying physics formula of the trajectory, but does’t know the relevant physics parameters of the objects and the environment. Predicting a trajectory directly from observations in this case is a nontrivial task as it requires the agent to understand the relationship between the strength of a shot and it’s trajectory. While in the real world physics parameters like gravity are (roughly) known, in game worlds these can and do have arbitrary values.
>
> “If I understand correctly, the trajectories (the input to InferNet) were generated with known G,V0, theta, and (the 3 latent variables of InferNet). It is not clear to me why the authors don’t use these for the MSE loss used to train InferNet (rather than using the projectile motion equation). “
>
>
> The goal of our paper was to avoid using the known values and to test the ability of the network to discover such values from observations. As described above, the motivation for this approach is that typically these values are unknown when one has no direct access to the “physics engine” (as it is the case in the real world and in some games).
>
>
> “- Is the ‘projectile motion equation’ missing from Fig.2-right; is it used for inference? Is G from InferNet also input to RelateNet?”
>
>
> The goal of the RelateNet is to learn to predict the two values V0 and theta directly from the given in-game variables such as relative position of the bird. In order to train RelateNet we use the values V0 and theta that were predicted by InferNet from the observed trajectories. G predicted by InferNet is used as input to the RelateNet.

---

### Decision · Program_Chairs · 2019-12-19

**Decision:**

Reject

**Comment:**

This paper aims to estimate the parameters of a projectile physical equation from a small number of trajectory observations in two computer games. The authors demonstrate that their method works, and that the learnt model generalises from one game to another. However, the reviewers had concerns about the simplicity of the tasks, the longer term value of the proposed method to the research community, and the writing of the paper. During the discussion period, the authors were able to address some of these questions, however many other points were left unanswered, and the authors did not modify the paper to reflect the reviewers’ feedback. Hence, in the current state this paper appears more suitable for a workshop rather than a conference, and I recommend rejection.